# Food Allergy Education and Management in Schools: A Scoping Review on Current Practices and Gaps

**DOI:** 10.3390/nu14040732

**Published:** 2022-02-09

**Authors:** Mae Jhelene L. Santos, Kaitlyn A. Merrill, Jennifer D. Gerdts, Moshe Ben-Shoshan, Jennifer L. P. Protudjer

**Affiliations:** 1Department of Food and Human Nutritional Sciences, Faculty of Agriculture, University of Manitoba, Winnipeg, MB R3T 2N2, Canada; santosmj@myumanitoba.ca; 2The Children’s Hospital Research Institute of Manitoba, Winnipeg, MB R3E 3P4, Canada; kmerrill@chrim.ca; 3Department of Biochemistry, Faculty of Science, University of Winnipeg, Winnipeg, MB R3B 2E9, Canada; 4Food Allergy Canada, Toronto, ON M2J 4A2, Canada; jgerdts@foodallergycanada.ca; 5Division of Pediatric Allergy and Clinical Immunology, Department of Pediatrics, McGill University Health Centre, Montreal, QC H4A 3J1, Canada; moshe.ben-shoshan@mcgill.ca; 6George and Fay Yee Centre for Healthcare Innovation, Winnipeg, MB R3E 0T6, Canada; 7Department of Pediatrics and Child Health, Rady Faculty of Health Sciences, College of Medicine, University of Manitoba, Winnipeg, MB R3E 0W2, Canada; 8The Center for Allergy Research Karolinska Institutet, 171 77 Stockholm, Sweden

**Keywords:** anaphylaxis, epinephrine, food allergy, schools, scoping review, teachers

## Abstract

Currently, no synthesis of in-school policies, practices and teachers and school staff’s food allergy-related knowledge exists. We aimed to conduct a scoping review on in-school food allergy management, and perceived gaps or barriers in these systems. We conducted a PRISMA-ScR-guided search for eligible English or French language articles from North America, Europe, or Australia published in OVID-MedLine, Scopus, and PsycINFO databases. Two reviewers screened 2010 articles’ titles/abstracts, with 77 full-text screened. Reviewers differed by language. Results were reported descriptively and thematically. We included 12 studies. Among teachers and school staff, food allergy experiences, training, and knowledge varied widely. Food allergy experience was reported in 10/12 studies (83.4%); 20.0–88.0% had received previous training (4/10 studies; 40.0%) and 43.0–72.2% never had training (2/10 studies; 20.0%). In-school policies including epinephrine auto-injector (EAI) and emergency anaphylaxis plans (EAP) were described in 5/12 studies (41.7%). Educational interventions (8/12 studies; 66.7%) increased participants’ knowledge, attitudes, beliefs, and confidence to manage food allergy and anaphylaxis vs. baseline. Teachers and school staff have more food allergy-related experiences than training and knowledge to manage emergencies. Mandatory, standardized training including EAI use and evaluation, and the provision of available EAI and EAPs may increase school staff emergency preparedness.

## 1. Introduction

Food allergy affects an estimated 7.0–8.0% of children worldwide, or about two children in an average-sized classroom of 25 children [1,2,3,4,5]. A food allergy is defined by Boyce et al. (2010) as “a potentially life-threatening immunological response that occurs reproducibly upon ingestion of the allergen” (p. 11) and has the potential to result in severe allergic reactions [6]. Anaphylaxis, the most severe type of allergic reactions, was operationalized by Sampson et al. (2006) as a “potentially fatal condition that involves multiple organ systems or, when exposed to a known allergen, low blood pressure” [7]. Anaphylaxis affects an estimated 2.0% of the North American population [6], with similar estimates (between 0.3% [8] to 3.1%) noted in European populations [9,10].

Prior to the coronavirus disease (2019-nCoV/COVID-19) pandemic, about 20.0% of anaphylactic reactions occurred in schools [11,12,13], an observation that is unsurprising given that children typically spend the majority of their waking hours at school. Most in-school reactions occurred in the classroom, cafeteria, and playground [13,14,15,16]. Of concern is that an estimated 30.0% of allergic reactions occurred among children who were not previously known to have a food allergy or had an allergy that was not communicated to school staff [13,16].

Currently, policies surrounding food allergy management and its implementation are diverse both across and within jurisdictions [17,18,19,20]. Recently, international recommendations on the prevention and management for childcare centers and schools [11] was published based on the available scientific literature. Authors noted the utility of the guidelines as “conditional”, wherein policymakers and stakeholders are to deliberate and adapt recommendations as needed to fit specific jurisdictional needs. Some of eight listed recommendations included school staff education and training, the removal of site-wide food bans and allergen-free zones, the requirement that children with a known food allergy had a current emergency anaphylaxis plan (EAP), and the availability of unassigned, or stock, epinephrine auto-injectors (EAI) in schools. Despite the need for further research in the topics described, this guideline may prompt jurisdictions to review and modify current policies.

The availability of EAI in school settings has been inconsistent. Students’ access to and carriage of prescribed EAI also varies [21], and by socioeconomic advantage [22]. Even when a student has an EAI, school policy may render access difficult if it is locked in an office or exclusively carried by a staff member [12,13,16,21]. In cases where a prescribed EAI was unavailable, almost half of students requiring emergency medication were treated with stock epinephrine [23,24]. Additionally, trained staff available to administer EAI are also diverse. When available, school nurses administer EAI [13,14,23,25,26]. That said, only 50.0% of nurses reported food allergy management training, of which 35.0% described being “self-taught” [26]. As school nurses may work part-time [21] and among several schools [27], distributed responsibility and training among other school staff who are at school premises at all times is warranted. In brief, policies addressing stock EAI and EAP implementation are underused despite key recommendations and available resources [10,11,27,28].

Despite the above-described variation in policy, management, and treatment, there is, to our knowledge, no previous synthesis of the extant literature on teachers and school staff’s knowledge and management practices of food allergy and anaphylaxis in schools. To this end, we aimed to conduct a scoping review on the in-school management of food allergies, and the perceived gaps or barriers in these management practices.

## 2. Materials and Methods

We performed a scoping review guided by the Preferred Reporting Items for Systematic Reviews and Meta-Analyses extension for Scoping Reviews (PRISMA-ScR) 2020 Checklist [29]. A literature search of original articles published in at least one of three medical literature databases (OVID-MedLine, Scopus, PsycINFO) was conducted on February 19, 2021. Search terms (see Appendix A) were identified in collaboration with content and methodological experts. Each search was filtered to child population and studies conducted in Canada, United States of America (USA), Australia, and Europe (including Turkey). Articles searched were restricted to publishing year 2006 and later to accommodate articles released subsequent to the implementation of Sabrina’s Law, a law passed in 2006 following the fatal anaphylactic reaction of 13-year-old Sabrina Shannon, in a school in Ontario, Canada. Sabrina’s Law requires every Ontario public school to implement an EAP for every student with food allergy including EAI administration instructions for staff [18].

Our primary outcome of interests were teacher and school staff management of food allergies in schools, including previous experience, knowledge and management of food allergy and anaphylaxis, emergency preparedness including availability of EAI and EAP, and school-based policies/guidelines. Studies were restricted to English and French. Additional inclusion criteria included previous experience in food allergy training, and experience working with students with food allergies, current practices, and food allergy knowledge of other school staff. There were no restrictions on type of study design. We excluded articles from grey literature, as well as abstracts, and publications without original data.

The search yielded 2010 articles (PsycInfo *n* = 61; Scopus *n* = 1414; OVID-MedLine *n* = 535). After the initial search and de-duplication (via Zotero *n* = 299; via Rayyan software [30] *n* = 10), there were 1701 articles, which were screened for titles and abstract by two independent reviewers (initials blinded for review; Figure 1). Titles/abstracts deemed potentially eligible for inclusion were advanced to full-text screening (*n* = 77). Full-text screening was made with consideration to study methods, participants, outcomes of interest, and findings. Full-text screening of English-language articles (*n* = 75) was conducted by two independent reviewers (initials blinded for review). French-language articles (*n* = 2) were full-text screened by a single reviewer (initials blinded for review) and excluded from the review. Two articles were reviewed by a third screener and were later excluded from the review [31,32].

As childcare centers may be housed in or proximate to schools, early learning and childcare centers were included in the initial search strategy. In the search strategy (Appendix A), childcare centers were termed “daycare” and “daycare centers” and “preschool” as per recommendations from the expert librarian. However, owing to the developmental differences of children in schools vs. childcare centers, we restricted the present review to schools only and thereafter excluded studies that had aggregate data on school and childcare centers’ teachers and staff. Data related to childcare centers will be reported elsewhere.

## 3. Results

From our search, 12 articles were included in the review, of which four (33.3%) studies were from North America [33,34,35,36] and eight (66.7%) [37,38,39,40,41,42,43,44] from Europe. About half of the studies (41.7%; 5/12) reported on teachers and school staff exclusively from primary school settings [34,37,38,39,42], 4/12 (33.3%) reported on mixed grade levels, the majority of which were primary schools [35,36,43,44], and 3/12 (25.0%) were presumed to represent primary schools [33,40,41] due to the language used, commonly differentiated in similar literature (e.g., “teachers” vs. “early childhood educators”) (Table 1). Most included studies did not have, or did not specify, any school food program participation (*n* = 10), or school nurse availability (*n* = 6). Two studies (16.7%) reported that its schools had school food programs [34,36], while four (33.3%) studies reported that some participating schools had a part-time nurse [34,35,36,39], and two (16.7%) studies reported that the Italian public school system had no school nurses available [43,44].

Overall, food allergy experience, training and education, baseline knowledge, and policies/ guidelines supporting food allergy management in schools were inconsistent between teachers and school staff, among and across jurisdictions.

### 3.1. Previous Experience in Food Allergy Management

The majority of teachers and school staff had experience with food allergies, as reported in 10/12 (83.3%) studies. However, higher proportions of teachers and school staff reported caring for a child with a food allergy compared to the teachers and school staff who had received training to do so.

An estimated 20.0–88.0% of Turkish, Italian, English, and American teachers and school staff reported having students with food allergies [33,37,38,42]. One study reported that 44.7% of Italian teachers had 1–2 students with a food allergy in their teaching experience, 31.6% had 3–5 students, and 23.7% had >5 students [42]. On average, United Kingdom (UK) schools enrolled between 1–12 students with a food allergy per school [37]. One Turkish study reported only 53.2% of participating teachers knew which students had a food allergy [39]. Fewer teachers (3.0–9.0%) reported they had taught students with a history of anaphylaxis than a food allergy [38,40]. Among UK schools, 57.0% (*n* = 89/157) reported having students who had previously had severe allergic reactions [37].

Rates of prior food allergy education were variable. Among Italian, Turkish, and Spanish teachers and school staff, rates of food allergy training ranged from 14.0–63.6% [38,41,42,43,44], whereas 43.0–72.2% of Italian [42] and Spanish [41] teachers and school staff reported no previous food allergy training at all. The majority of Italian and Turkish teachers (71.7–82.3%) reported having first aid training, although the extent of food allergy training included (e.g., EAI administration) was unspecified [38,43,44]. In Washington state, USA, approximately half (51.1%; 1102/2156) of teachers reported previous food allergy training. Of these same teachers, 62 reported having administered an EAI, although not all (77.4%; 48/62) had prior EAI training [35].

The method of food allergy education delivery was reported in 40.0% (4/10) of studies. Italian, Turkish, and American teachers and various school staff received previous food allergy education primarily from first aid courses (71.7%) [34,43], health training (11.1%) [43], mass media (22.4–64.5%) [39,43], the internet (17.9–23.0%) [39,43], booklets (37.3%) [39], seminars (22.4%) [39], and less commonly, via acquaintances or relatives (1.4%) [43]. Other sources of food allergy information included sessions from in-service days and/or regional conferences [39], parents, and individuals with a food allergy [42].

### 3.2. Baseline Knowledge

Teachers and school staff reported poor knowledge of food allergy understanding and anaphylaxis management at baseline in 6/12 (50.0%) studies from Italy, Spain, USA, and Turkey [34,38,39,40,42,43]. Turkish and Italian teachers and school staff had knowledge of allergic reaction symptoms, but a poor understanding of food allergy and anaphylaxis management [39,42,43]. Notably, Italian teachers and principals from primary schools had statistically significant higher baseline questionnaire scores than middle schools (*p* < 0.001) when compared through one way analysis of variance and Bonferroni post hoc test [43].

The majority of American primary school teachers from economically-advantaged and disadvantaged areas (78.3% and 76.5%, respectively) [34] and one group of Italian teachers and school staff of various grade levels (79.3%) were able to identify common allergenic foods [43], compared to approximately 40.0% of Turkish primary school teachers, and another group of Italian primary school teachers, who correctly answered questions about food as allergic triggers [39,42]. Interestingly, one group of primary school Italian teachers acknowledged having poor food allergy knowledge (mean = 5.1/10; Standard Deviation (SD) = 2.1) but perceived food allergy as a significant issue in schools (mean = 7.6/10, SD = 2.1, based on a scale of 1–10, with higher scores corresponding to higher significance) [42].

The economic advantage of school areas appeared to also influence teacher and school staff’s baseline food allergy knowledge. Primary school teachers in both Houston, USA, and Turkey from schools in economically-advantaged areas had more non-statistically significant higher baseline food allergy knowledge than teachers from economically-disadvantaged areas [34,39].

Anaphylaxis knowledge was likewise poor as reported in 3/12 (25.0%) of studies from Italy, Spain, and the USA [34,40,43]. Italian and Spanish authors determined that teachers’ and school staff’s baseline knowledge was not influenced by previous education on food allergy or experience working with students with an anaphylaxis history [40,43].

An estimated 45.3% of Spanish primary school teachers [40] and 65.4% of Italian [43] teachers of various grade levels correctly reported that epinephrine is the main anaphylaxis treatment. Similar rates of anaphylaxis treatment knowledge were reported by American teachers from economically-disadvantaged areas (45.3–49.0%) compared to teachers from economically-advantaged areas (70.0–80.6%) [34]. Conversely, fewer Italian teachers and principals of various grade levels (34.5%) knew epinephrine was safe to use for suspected anaphylaxis without severe side effects [43]. Fewer Spanish and Turkish primary school teachers and canteen staff knew what an EAI was (10.1% [39], and 18.9% [40], respectively), or how to use an EAI (6.8–13.2%) [39,40] and where to administer it (3.8%) [39]. If faced with a food allergy-related emergency, only 24.5% of Turkish primary school teachers stated they would administer first aid, although none of the teachers identified that epinephrine was the appropriate medication to use [39].

### 3.3. In-School Emergency Preparedness

Food allergy-related emergency preparedness, with regard to self-efficacy, confidence, and food allergy-related emotions, was discussed in 6/12 (50.0%) of studies, all of which were European [37,38,41,42,43,44].

Self-efficacy in managing food allergies in school was discussed in three studies, all of which made use of the School Personnel’s Self-Efficacy in Managing Food Allergy and Anaphylaxis (SPSMFAA) questionnaire by Polloni et al. (2016) [32] to measure self-efficacy on food allergy management. The questionnaire measures a total of 40 points based on eight factors (1 = cannot do, 5 = highly certain can do) [32]. Compared to anaphylaxis management, food allergy management was associated with greater self-efficacy [38,41,44]. Turkish primary school teachers exhibited that previous food allergy experience and food allergy training were associated with greater self-efficacy in managing a food allergy and anaphylaxis (*p* < 0.001) [38]. In fact, significant SPSMFAA score differences were seen among Turkish primary school teachers with previous food allergy training compared to those who did not have previous training (mean = 26.74/40 ± 6.21, vs. 22.18/40 ± 7.48, respectively; *p* < 0.001) [38].

Confidence in managing anaphylaxis was reported by approximately half (47.3%; 53/112) of UK primary schools, with no difference (*p* = 0.10) among schools with or without students with a food allergy (52.6% vs. 36.1%, respectively) [37]. Most UK schools (60.7%) demonstrated being prepared for allergic reactions in students without a previous allergic history by establishing communication and documentation systems, and identifying staff member roles in the event of an allergic emergency, with no significant difference between schools with vs. without students with food allergy enrolled (61.0% vs. 60.0%, respectively; *p* = 0.94) [37].

Elsewhere, Italian teachers and principals of various grades reported food allergy-related emotions were concern (66.9%), anxiety (15.8%), fear (3.7%), and helplessness (7.0%). Positive attitudes were also associated (9.3%) in relation to newfound post-intervention knowledge [43].

Three focus groups of Italian primary school teachers (*n* = 25) qualitatively discussed concerns over managing the child in crisis and other students in class [42]. Teachers were unauthorized to administer certain (unspecified) drugs, thus, had restricted emergency management abilities to providing first aid and calling for help. It was not disclosed what type of first aid treatment teachers were allowed to perform. Feelings of insecurity were described, and teachers felt unable to manage emergencies due to the perceived lack of food allergy knowledge. Additionally, teachers thought that the responsibility of food allergy management was beyond their teaching duties and required more emotional involvement [42].

### 3.4. School-Based Policies and Guidelines

School-based policies/guidelines were described in 5/12 (41.7%) of studies, although implementation and adherence were variably enforced among participating schools [34,36,37,39,40]. An outline of policies and guidelines are listed in Table 2.

EAP usage was inconsistently implemented (5.9–89.5%) among participating schools from Italy, UK, and the USA [36,37,39,40]. EAI was available, as prescribed in one UK study [37,42], and unspecified in one Spanish study [40]. In Spanish schools where EAI was available, only 66.0% of teachers and school staff reported to know where it was located [40]. One Houston, USA-based study stated more stock EAI was available in two schools in economically-advantaged areas (*n* = 6–9 per school) compared to two schools in economically-disadvantaged areas (*n* = 1 each) [34].

Food bans and mealtime accommodations were the most common policies imposed in schools as reported by 3/12 (25.0%) of the Milwaukee, USA; Spanish; UK studies [36,37,40]. Other preventative policies implemented among these schools were distancing measures, e.g., separate lunch table for students with food allergies, safe food/utensil handling, handwashing, surface cleaning, food sharing, and reviewed food items for classroom projects [36,37]. Teachers were primarily responsible for carrying out tasks to manage food allergies such as mealtime supervision [36,37] and meeting with the parents and students with food allergies [40].

In a study by Eldredge et al. (2014), of which 76.1% of responding Milwaukee schools included primary school students, the authors reported on rates of food allergy policy implementation. Authors also noted that policies in this school district were independently determined by governing parishes and/or school boards. Nevertheless, enrollment of students with food allergy appeared to determine policy/ guideline implementation. In this study, 71.0% (53/75) of schools reported some policy/guideline in place. Schools with students with food allergies had an increased likelihood of implementing policies compared to schools without students with a food allergy (Odds Ratio (OR) = 6.30, 1.50–2.60). In fact, 85.0% of schools who had students with a food allergy enrolled had policies implemented, compared to the 15.0% of schools without policies (*p* ≤ 0.0001). Schools with policies were also 3.5 times more likely to require EAPs than schools without policies (67.0% vs. 35.0%, respectively; *p* < 0.0001; OR = 3.50, 95% Confidence Interval (CI) = 1.00–12.20) [36].

In a UK study of primary schools, 76.0% (111/152; 95% CI = 68.0–83.0%) reported having a standard management protocol. An estimated 0.7% (165/24,174) of students had a history of anaphylaxis, or were at risk for severe reactions, and had an EAI. Compared to schools at which there were no students at risk for anaphylaxis, schools attended by students at risk were significantly more likely to have a standard management protocol (57.0% vs. 90.0%, respectively; *p* < 0.001) [37].

### 3.5. Post-Educational Intervention Knowledge

Interventional education sessions were described in 8/12 (66.7%) of studies. Sessions were delivered through a healthcare provider-led presentation. One-third (4/12; 33.3%) of studies also provided hands-on EAI training [35,40,41,44].

Overall, teachers and school staff who received interventional education demonstrated better knowledge on food allergy and anaphylaxis management [33,34,35,40,41,42,43,44] compared to their baseline knowledge or versus controls [33,34]. The key outcomes of each study are listed in Table 3.

Sustained knowledge and confidence levels were also described in one American longitudinal study that followed-up with participants, including teachers and school staff from various grade levels, 3–12 months post-intervention. Participants reported sustained confidence levels in the recognition of signs and symptoms, ability to prevent food allergic reactions, and knowing what to do during an anaphylaxis emergency [35]. Primary key messages recalled by 57.0% of participants 3–12 months post-intervention included EAI administration, reaction signs and symptoms, importance of following an EAP, and providing immediate treatment [35]. A small proportion of participants (*n* = 22) experienced a food allergy emergency post-intervention, 42.8% of which were caused by unknown allergens and 23.8% occurred in primary schools. Of these participants, 81.8% (18/22) had previous training before the study intervention. Nevertheless, 61.9% found that the recognition of food allergic signs and symptoms and 52.3% found the hands-on EAI training useful in real-life situations [35].

In a Houston, USA-based study, the intervention group teachers from economically-disadvantaged school areas had non-significant higher questionnaire scores post-intervention than teachers from economically-advantaged schools in both intervention and control groups [34]. Another Houston study that compared teachers who received intervention to those who did not, reported that there was no correlation between level of education (<4 years college, 4 years college, and graduate degree) and the survey scores [33]. Spanish primary school teachers and school staff exhibited significantly better food allergy knowledge (*p* < 0.001) through improved recognition of anaphylaxis (40.0% to 81.0%, respectively), knowledge about when (19.0% vs. 100.0%, respectively) and how (13.0% vs. 100.0%, respectively) to use an EAI, albeit authors reported modifying acceptable questionnaire responses as the original questions were “not easy to answer” [40]. Education sessions were deemed useful by Italian primary school teachers (8.6/10 ± 1.67; on a scale of 1–10, where 10 = very useful) [42]. Another group of Italian teachers and principals from various grade levels showed significantly better questionnaire scores post-intervention (mean = 6.6/10 vs. 8.9/10, respectively; *p* < 0.001) [43]. Post-education, the same Italian group of teachers and principals agreed anaphylaxis is manageable at school (82.6% vs. 96.5%, respectively; *p* < 0.001) and school staff are responsible for food allergy management (82.8% vs. 93.9%, respectively; *p* < 0.001) [43].

Interventional education influenced teachers and school staff’s beliefs and attitudes about food allergy management. Among Houston, USA-based private school teachers, those in the intervention group, compared to control group teachers who did not receive intervention, tended to show greater agreement about the importance of EAI as a lifesaving measure for anaphylaxis. Although the authors identified an OR = 873.77 (*p* = 0.173), the difference was statistically insignificantly different because, as the authors noted, “almost all” participants agreed or strongly agreed with the importance of EAI [33]. Similarly, compared to the baseline, intervention group teachers were 3.3 times more likely to recognize the seriousness of food allergies (OR = 3.30; 95% CI = 1.60–6.70; *p* = 0.001) and to agree that students with food allergies are likely to experience discrimination (OR = 3.30; 95% CI = 2.00–5.50; *p* = 0.01) [33]. Intervention teachers were also 52 times (OR = 52.0; 95% CI = 2.90–930.75; *p* < 0.01) more aware, post-intervention, that students with food allergies experienced bullying compared to control teachers, with 26 times increased likelihood of agreement that students with food allergies experienced bullying (OR = 25.55; 95% CI = 9.86–66.25; *p* < 0.001) [33].

Education sessions were associated with increased confidence [35], comfort level [34], and self-efficacy [41,44] in the majority of participants, regardless of whether participants had previous training [35,41,42,44]. The majority of American participants (>94.0%), some of whom were teachers and school staff, answered opinion statements positively post-intervention, indicating more confidence in prevention, recognition, and response skills to food allergy emergencies [35]. Significant post-intervention SPSMFAA scores [32] were reported for Spanish teachers and school staff (*p* < 0.05) in food allergy management items, specifically in putting an EAP in place for students with a food allergy, managing students at risk of reactions to food, and recognizing anaphylaxis symptoms and administering EAI in anaphylaxis management [41]. Following a food allergy intervention, Italian teachers’ and school staff’s post-intervention scores were higher compared to pre-intervention studies. The greatest differences were seen among those with low self-efficacy at baseline [44].

### 3.6. Future Educational Needs

The majority of primary school teachers and staff expressed an interest in receiving more food allergy and anaphylaxis training [36,37,39,42]. Teachers also thought that increasing food allergy awareness in schools and involving all students may increase empathy among all schoolchildren [42]. To deliver further food allergy education and awareness, study participants suggested establishing online repositories for educational resources, have more in-person training or video training [36,42], and have students with food allergies wear medical alert accessories to inform others of their condition [39]. Additionally, nearly all (94.0%) of UK teachers either “agreed” or “strongly agreed” that unprescribed EAI ought to be kept in schools [37]. Interestingly, schools with no students at risk of anaphylaxis were non-statistically significantly more likely to agree than schools with students at risk of anaphylaxis (55.6% vs. 30.3%, respectively; *p* = 0.09) [37].

## 4. Discussion

In this scoping review of the European and North American literature on in-school management of food allergies, we identified several perceived gaps and barriers in management. First, teachers and school staff acknowledged the significance of food allergies [42] yet lacked experience and knowledge. We identified participants’ knowledge differences [33,39] and EAI availability [34] from schools in economically-advantaged and disadvantaged areas. Studies also reported that teachers and school staff did not know which students had a food allergy [37,39]. Second, there exists wide variation, and reporting, of food allergy management practices including the provision of policies/guidelines, EAP implementation, and inconsistency in EAI availability and knowledge in EAI administration, as similarly described in other studies [13,14,22,24,25]. Third, preparedness and self-efficacy of teachers and school staff to manage anaphylaxis effectively are correspondingly variable. Unsurprisingly, additional training was desired by many.

The need for additional training is underscored by the commonality of students with food allergies, juxtaposed against inconsistent policies across and between jurisdictions [17,18,19,20]. As school staff are likely to be the first adults to be notified of food allergy-related emergencies [15], adequate and universal emergency management skills are essential in student safety, including EAI administration. One USA-based study in our review reported that not all teachers have administered EAI but have not been previously trained [35], which illustrated that teachers are key players in emergency management in schools, especially when there are no school nurses available. School nurses have also reported to have inconsistent training, or were “self-taught” [15,25,26]. Reliance on one nurse to manage medical emergencies may be impractical as allergic reactions can occur anywhere within school premises. Additionally, if parents are less involved and/or unaware of serious food allergy concerns, teachers may also assume caregiving responsibilities and help students learn about their own food allergy management.

Our review highlights the need to share food allergy management responsibilities, including, but not limited to, maintaining individual EAPs, knowing where EAI are located and how to use it, promoting preventative practices (e.g., handwashing) and recognizing signs and symptoms of allergic reactions, and knowing own roles in emergencies by providing food allergy training for all teachers and school staff, including school nurses where available. Such training may also reduce the propensity of other school staff to turn to online, non-academic resources for food allergy education [39,42,43]. Moreover, early (pre-hospital) treatment decreases the risk of hospitalization [13], while delayed treatment from symptom onset was associated with the risk of having a biphasic reaction and fatality [12,24]. As the long-term effects of staff training on food allergy management knowledge are unknown, the implementation of post-training evaluation may also be beneficial [11].

School meal programs also raise the value of food allergy training for other school staff such as cafeteria personnel and food monitors, as proper food handling and preparation are foundational in preventing allergic reactions [6,46]. Our study reported on two studies with school food program participation that did not discuss how food allergies were accommodated [34,36]. Future training programs should also address how school food programs apply food allergy education in practice, including safe food handling training, cleaning protocols, and increased mealtime supervision for younger students who may have more impulsive behaviors [47].

Although a universally accepted EAP and laws to provide stock epinephrine in schools would prove challenging to develop and garner acceptance, we purport that such calls are essential at a national, or regional level. Collaborative efforts and partnerships among all stakeholders including affected students and families should focus on identifying students at risk of anaphylaxis. Thus, planning and implementation of medically sound EAPs, yet relevant and clearly understood by its intended users, is essential. Additionally, in conjunction with staff training and the implementation of EAPs, stock EAI in schools would be advantageous as not all students with a food allergy may have an EAI, or do not carry them around school. Meetings with teachers, children, and their families may also increase communication and consensus on stock EAI usage and care plans [40]. Likewise, training, EAP implementation, and stock epinephrine availability align with international recommendations [11,28], and may increase staff awareness of food allergies, and help alleviate concern, anxiety, fear, and helplessness reported by teachers and school staff [43]. In turn, training may contribute to teachers and school staff’s confidence, self-efficacy, knowledge, and ability to perform in emergency situations.

To our knowledge, this is the first scoping review to provide an overview in some school jurisdictions in Europe and USA. We did not restrict the publications to the English language only and presented available data from multiple Western countries. Our review also extends the findings from Waserman et al. (2021), such as the positive uptake and perceived benefits of teachers and school staff of food allergy training, providing available EAI and implementation of action plans [11].

We acknowledge that searching only within three databases and the publication year cut-off may have introduced some reporting bias and reduced eligible studies. We also did not perform a quality appraisal of the included studies or comparisons of the interventions. Moreover, our ability to compare the interventions and results into a cohesive analysis were limited given the heterogeneity of design of the included studies [48]. However, we were able to identify common themes. We recognize that we excluded all grey literature, as well as publications outside Europe and North America, and in languages other than English or French.

Nevertheless, our review highlights several key take-away messages (Box 1), including the need for further research and the creation of a food allergy training strategy that includes EAI administration for all school staff. Our review findings can also be used to inform policymakers to consider implementing an evaluation program for existing training courses. In light of the COVID-19 pandemic, the usage of virtual platforms for training purposes can be an accessible communication medium. Lastly, the provision of stock EAI and individualized EAPs should be considered as mandatory as jurisdictions are able. The execution of such may pose greater benefits beyond having available rescue medication but may also help increase the confidence and self-efficacy of teachers and staff to be able to manage emergency situations appropriately.

Box 1Key take-away messages.
Teachers and school staff play a pivotal role in emergency response. At baseline, teachers and school staff have poor and variable knowledge and experience of food allergy.Teachers and school staff may benefit from standardized, annual food allergy training.Key elements of food allergy training may include epinephrine auto-injector (EAI) administration, causal foods, signs and symptoms of a reaction, and importance and usage of a emergency anaphylaxis plans (EAP).Implementation of EAP for all students with a food allergy and having stock EAI, in conjunction with annual training will improve student safety and schools’ emergency preparedness.


## 5. Conclusions

In conclusion, current in-school management of food allergies, including food allergy education, are highly heterogeneous across jurisdictions in western nations for which data are available.

Implementation, continuation and/or evaluation of universal standardized training, usage of personalized EAPs, provision of stock EAI in schools, and policy or guideline implementation outlining these practices may be considered by schools and governing jurisdictions. As such, these actions will support teachers and staff in preventing and managing in-school food allergy emergencies safely and effectively.

## Figures and Tables

**Figure 1 nutrients-14-00732-f001:**
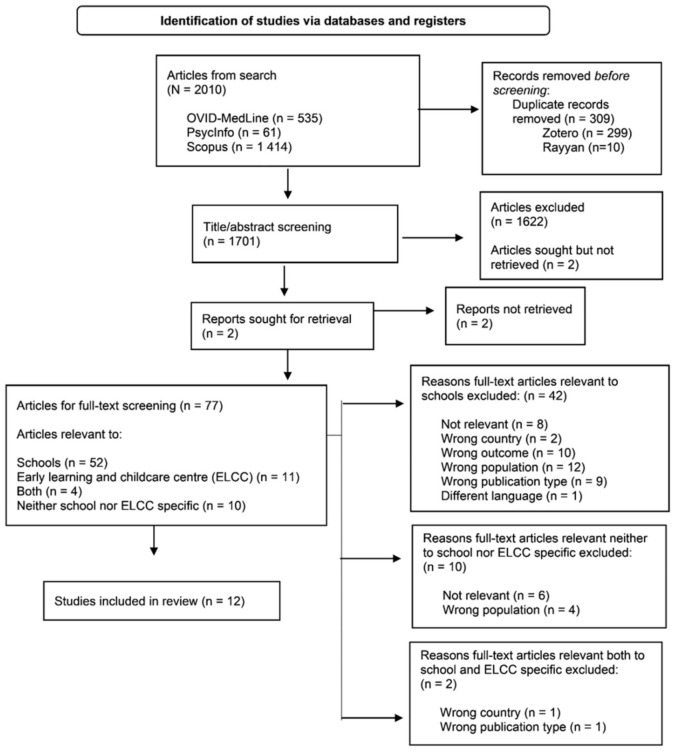
PRISMA flow diagram depicting the selection process articles and reports in the current scoping review.

**Table 1 nutrients-14-00732-t001:** Summary of articles’ country of origin, research design, methods, and population, presented in alphabetical order by first author’s last name.

First Author, Year	Country	Research Design	Methods	Teachers and School Staff (*n*)	Type of School (*n*)
Polloni 2013 [43]	Italy	Quasi experimental pre/post-intervention	School staff attended an educational course by the Veneto Food Allergy Center and completed pre/post surveys.	1184 Teachers and Principals	Primary school (*n* = 598) Middle and high school (*n* = 291)
Polloni 2020 [44]	Italy	Quasi experimental pre/post-intervention	Teachers and school caretakers (class assistants and meal supervisors) participated in an educational intervention by the Veneto Food Allergy Center. The SPSMFAA questionnaire [32] was completed pre/post-session.	592 Teachers (*n* = 474) Caretakers (*n* = 118)	Primary school (*n* = 216) Middle and high school (*n* = 152)
Ravarotto 2014 [42]	Italy	Mixed methods (Focus group, pre/post-intervention)	Phase 1: 3–90-minute focus groups of teachers informed the intervention’s communication strategy. Phase 2: Information workshop and “*The Theatre of Health*” show was held in various provinces. Phase 3: Teachers who attended the session completed pre/post questionnaires.	Three focus groups (*n* = 25 participants) Information workshop (*n* = 197) Assessment questionnaires (*n* = 158)	All primary schools. Focus groups (*n* = 3) Information workshops and questionnaire (*n* = 5)
Gonzalez-Mancebo 2019 [41]	Spain	Quasi experimental pre/post-intervention	“Management of Food Allergy in Children and Adolescents in School Centers” conference participants were provided an education session and a pre/post SPSMFAA questionnaire [32]. Training efficacy results between cafeteria monitors and teachers were compared.	191 Cafeteria monitors (*n* = 97) Teachers (*n* = 46) Cooks (*n* = 25); Other professions (*n* =23)	Number of primary schools not reported
Rodríguez Ferran 2020 [40]	Spain	Multi-center quasi experimental pre/post-intervention	Teachers and canteen staff from three schools, as requested by patients’ family members, participated in an educational session and pre/post questionnaire. Grade-specific data were not disclosed.	53 Teachers (*n* = 45) Canteen staff (*n* = 8)	Varied types of schools included. (*n* = 3) Schools had students aged 3–12y.
Ercan 2012 [39]	Turkey	Cross-sectional survey	Private and public-school teachers completed questionnaires, and food allergy knowledge was compared.	237 Public school teachers (*n* = 91) Private school teachers (*n* = 146)	Number of primary schools not reported
Ozturk Haney 2019 [38]	Turkey	Cross-sectional survey	Private and public-school teachers participated and completed the SPSMFAA questionnaire [32].	282 Public school teachers (*n* = 169) Private school teachers (*n* = 113)	All primary school (*n* = 12), of which 4 were private and 8 were public.
Canon 2019 [33]	USA	Multi-center pre/post-randomized intervention	Six Houston private schools were assigned to intervention (*n* = 4) or control groups (*n* = 2). Both groups completed the Chicago Food Allergy Research Survey [45]. Intervention groups received education sessions while control groups did not, and food allergy knowledge was compared.	375 Intervention (*n* = 302) Control (*n* = 73)	All private schools (*n* = 6)
Eldredge 2014 [36]	USA	Cross-sectional survey	Private, parochial schools participated in the survey. Electronic questionnaires were answered by principals or administrators. Grade-specific data were not disclosed.	78 Principals (*n* = 70) Administrators (*n* = 8)	Varied types of schools included. (*n* = 71) 76.0% were pre-K/K-6th or 8th grade.
Shah 2013 [34]	USA	Multi-center pre/post- randomized intervention	One school each from higher/ lower socioeconomic areas in the Houston area were recruited. Intervention groups received education sessions while control groups did not, and food allergy knowledge was compared.	Pre-intervention (*n* = 195) Post-intervention (*n* = 131)	All public primary schools (*n* = 4)
Wahl 2015 [35]	USA	Quasi experimental pre/post- intervention	A school and community personnel training program provided education sessions and a survey. A follow-up survey was given 3–12-months post-intervention. Participants who participated in a food allergy emergency post-intervention were followed-up via phone interviews.	Primary survey (*n* = 4088) Secondary survey (*n* = 332) Phone interview (*n* = 21) Participant roles: Teachers (48%) Childcare providers (6%) School Aide (5%) Administrator (5%) School Nurses (2%) Other (34%) (Included camp counsellors, bus drivers, multiple of specified job titles, parents, volunteers, coaches, food service workers or no indication of job title)	Varied types of schools included. Number of primary schools not reported.
Raptis 2020 [37]	UK	Cross-sectional survey	All schools in the region were invited to participate in the survey. Only primary school data was presented in this study.	Specific participant roles not reported.	Primary schools (*n* = 157) High schools (*n* = 22) *

Abbreviations: *EAI* = epinephrine auto-injector; *K* = Kindergarten; *NS* = not specified; *SPSMFAA* = School Personnel’s Self-efficacy in Managing Food Allergy and Anaphylaxis; *UK* = United Kingdom; *USA* = United States of America; *y =* years. * High school data were excluded in the paper per author reports.

**Table 2 nutrients-14-00732-t002:** Summary of in-school policies, emergency action plan, epinephrine auto-injector availability, and other management practices among schools, presented in alphabetical order by first author’s last name.

First Author, Year	Policies	EAP Availability	EAI Availability	Other Management Practices
Eldredge 2014 [36]	71.0% of schools had some sort of guideline/policy for food allergy while 25.0% of schools had none.	56.0% of schools required an EAP.	Not reported	76.0% of schools needed special arrangements (i.e., peanut-free classroom, allergen-free areas or cafeteria tables, increased monitoring, physical distancing, and having special meals for students with food allergy). 57.0% of schools had handwashing guidelines. 30.0% had no food sharing policies. 58.0% had classroom project food substitution guidelines and 45.0% had cleaning surfaces with allergen contact.
Ercan 2012 [39]	Not reported	6.0% of teachers, all from private schools, had available EAP. 86.0% of teachers had no EAP, and 8.0% were uncertain if EAPs were available.	Not reported	Not reported
Raptis 2020 [37]	76.0% of schools had standard protocols related to allergic reactions.	89.5% of schools reported having an EAP for students with anaphylaxis history.	0.7% (*n* = 165) of students with food allergy had prescribed EAI. 45.2% of schools reported their students at risk of anaphylaxis carried an EAI.	Schools had guidelines for: staff food handling guidelines (79.0%), special mealtime supervision (49.0%), no food sharing policy (63.0%), no utensil sharing policy (45.0%), aware of food packaging regulations (66.0%), reviewed curriculum to remove allergen foods (68.0%), and no eating on transportation policy (48.0%), communication systems during emergencies (94.1%), identifying staff roles (82.1%), documenting staff emergency response (81.9%), and preparing for allergic reactions in students without prior allergic history (60.7%).
Rodriguez Ferran 2020 [40]	Not reported	83.0% of teachers and school staff reported they had EAP.	66.0% of teachers and school staff knew where EAI was in their school.	56.0% of teachers and school staff had meetings with parents/guardians of students with food allergy in their care.
Shah 2013 [34]	Not reported	Not reported	Schools in economically-disadvantaged areas had 1 EAI each. Schools in economically-advantaged areas had 6 and 9 EAI each.	Not reported

Abbreviations: *EAI* = epinephrine auto-injector; *K* = Kindergarten; *NS* = not specified; *SPSMFAA* = School Personnel’s Self-efficacy in Managing Food Allergy and Anaphylaxis; *UK* = United Kingdom; *USA* = United States of America; *y* = years.

**Table 3 nutrients-14-00732-t003:** Summary of studies that provided educational interventions (*n* = 8), presented in alphabetical order by first author’s last name.

**First Author, Year,** **Country**	**Intervention and Session Topics**	**Key Intervention Outcomes**
Canon 2019 [33] USA	1-hour education session with HCP Sessions taught case scenarios, common food allergens, routes of exposure, reaction recognition and prevention, epinephrine administration, importance of EAP, bullying of students with food allergy and classroom protocols.	Intervention group had higher post-intervention survey scores compared to controls (95% CI = 16.62–22.53; *p* < 0.001) and their pre-test surveys (95% CI = 18.17–21.38; *p* < 0.001). Intervention vs control group post-intervention were more likely to recognizing food allergy as life-threatening and agree that children with food allergy were treated differently and bullied (*p* < 0.001), 5 times more likely to acknowledge food avoidance is hard (*p* = 0.003) and 874 times more likely to agree that EAI is an important lifesaving measure and use it in an emergency (*p* = 0.173).
Gonzalez-Mancebo 2019 [41] Spain	Education session and EAI workshop for school staff included practical EAI training. Sessions taught food allergy definition, diagnosis, problems of children with food allergy in school settings, allergic reaction recognition and prevention measures, coordination of care, anaphylaxis treatment and, and EAP discussion	Significant improvements in SPSMFAA questionnaire [32] mean scores were observed (*p* < 0.05). The largest pre-post mean score difference was in managing allergen avoidance (mean = 4.29, SD = 0.98 vs. mean = 4.51, SD = 0.72). The smallest difference was in administering drugs (e.g., EAI) to a student having a severe and sudden reaction (mean = 3.08, SD = 1.41 vs. mean = 4.51, SD = 0.84). Case study scores also improved from pre- post intervention (25.5% vs 96.9%, respectively).
Polloni 2013 [43] Italy	2-hour session with a pediatric allergist, dietician, psychologist, and a lawyer. Session topics were not specified.	Primary school teachers scored higher than nursery or high schools (F-value: 13.450, df = 2, *p* < 0.001). Mean scores significantly increased from pre-post-intervention. From pre-post-intervention, more participants thought anaphylaxis could be managed in schools (82.6% vs. 96.5%, respectively; *p* < 0.001) and is school staff responsibility (82.8% vs. 93.9%, respectively; *p* < 0.001). Feelings related to food allergy management were concern (66.9%), anxiety (15.8%), fear (3.7%) and helplessness (7.0%).
Polloni 2020 [44] Italy	2-hour session with an allergist, psychologist, and a lawyer. Practical EAI training was included. Sessions taught description of allergic mechanisms, signs and symptoms, prevention and treatment, explanation of EAPs and presentation of national and regional regulations on food allergy-related drug administration in schools and discussions on food allergy-related psychosocial and emotional issues.	Improvements in SPSMFAA questionnaire [32] mean scores were observed. Post–pre score differences in anaphylaxis management (0.67–1.67, respectively), was higher than food allergy management difference (0.2–1.0, respectively). The largest pre-post mean SPSMFAA [32] score difference was in administering drugs (e.g., EAI) to a student having a severe and sudden reaction (mean = 1.3) and the lowest in guaranteeing students with food allergy full participation to all school activities (mean = 0.47). Median scores increased, as evaluated through conditional regression, from pre-post-intervention (<17 to 25, respectively), independent of all other covariates (type of job, age, school, gender, previous anaphylaxis and food allergy knowledge, training, and experience).
Ravarotto 2014 [42] Italy	2-hour workshop with allergist or pediatrician, a veterinarian, and a scientific communication expert. Sessions taught common allergenic foods, difference between allergy and intolerance, allergic reaction signs and symptoms, first aid introduction, available training tools/ resources and regulations to protect consumers	The number of correct answers determined knowledge categories. Pre-intervention, 3.2% had poor knowledge, 56.3% had fair, 39.9% had satisfactory, and 0.6% had good knowledge. Post-intervention, the percentage of correct answers increased to 1.3% fair, 67.7% satisfactory, and 31.0% good knowledge. Increased knowledge was unrelated to previous food allergy training (χ2 = 0.143, *p* = 0.931).
Rodríguez 2020 [40] Spain	40–50-minute presentation by pediatric allergist and a 10–20-minute EAI practical session by pediatric nurse. Sessions taught allergy definition allergic reactions pathophysiology, reactions prevention and recognition, communication with family and EAP development, anaphylaxis management, legal aspects and official recommendations.	From pre-post-intervention, participants had significantly better anaphylaxis recognition (40.0% vs. 81.0%, respectively; *p* < 0.001). Knowledge of how and when to use the EAI increased from 19.0% and 13.0%, respectively, to 100.0% of participants (*p* < 0.001).
Shah 2013 [34] USA	1-hour education session with physician. Sessions taught food allergy prevalence, causal foods, signs of local and systemic reactions, reaction prevention and treatment.	Teachers in the economically-disadvantaged vs. economically-advantaged school areas had a larger increase in correct answers post-intervention (34.6%; 95% CI = 32.1–103.9 vs. 24.6%, 95% CI = 21.5–74.1, respectively). Teachers from both economically-disadvantaged and advantaged school areas had increased scores from pre-post-intervention in questions related to treatment of local and systemic reactions, causal foods, and signs of anaphylaxis.
Wahl 2015 [35] USA	45-minute presentation by a food allergy Educator nurse. Practical EAI training was included. Sessions taught key food allergies facts, allergic reactions, prevention, and recognition, and importance of immediate treatment.	Post-intervention, most teachers and school staff had better confidence in prevention of allergic reactions (94.0%), recognizing reaction signs and symptoms (96%), know what to do in an emergency (97%), and administer an EAI (94%). Approximately half of participants had prior food allergy training. 95.0% of participants had positive feedback about food allergy management confidence in preventing allergic reactions, symptom recognition, and knowing what to do in emergencies 3–12-months post-intervention. 57.0% of participants recalled three key messages from the sessions. 21 participants who experienced a food allergy emergency post-intervention were interviewed. 61.9% found that signs and symptoms recognition and 52.3% reported EAI training were useful in real-life situations.

Abbreviations: *EAI* = epinephrine auto-injector; *HCP* = healthcare professional; *UK* = United Kingdom; *USA* = United States of America.

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
