# Peer review of "Food Allergy Education and Management in Schools: A Scoping Review on Current Practices and Gaps"

_nutrients, 2022, doi:10.3390/nu14040732_

Round 1

Reviewer 1 Report

The topic presented by the authors is very important for the health of children at school. The review is very well written and includes the important literature. The conclusions drawn are an important message.

Minor changes required: Please give the statistical parameters in italics.

Author Response

Nutrients 1573383

Due date for revisions: 20220131

We wish to thank the reviewers and editors for the careful review of our recently submitted manuscript, titled Food allergy education and management in schools: A scoping review on current practices & gaps. These comments have contributed to an improved manuscript, and to which we provide point-by-point responses below. The line numbers in the below correspond to those in the clean manuscript. We remain optimistic that these improvements are satisfactory to the reviewers and that the manuscript will be accepted for publication.

REVIEWER ONE

Comment 1

The topic presented by the authors is very important for the health of children at school. The review is very well written and includes the important literature. The conclusions drawn are an important message.

Response to Comment 1

Thank you for taking the time to review our paper and for the kind comments.

Comment 2

Minor changes required: Please give the statistical parameters in italics.

Response to Comment 2

Thank you for pointing that out. All statistical parameters have been italicized per reviewer recommendations. Please see the changes on lines 177, 200, 226, 228-229, 236. 277, 279, 281, 304, 311-313, 319, 323-325, 328, 335 and 365.

Reviewer 2 Report

 This was an interesting review that explores in-school management of food allergy, and perceived gaps or barriers in these management practices. The manuscript is certainly unique in its own right, however, there one major concern worth raising.

My primary concern with this review involves the Methods section.

Firstly, the use of additional electronic academic databases would likely have added to the study in both complexity and sample size. If the scope of this study were expanded to use additional databases, more sources might have been identified and explored. For example, the following electronic databases could have been used for a more thorough and inclusive search;

“ArticleFirst; Biomed Central; BioOne; BIOSIS; CINAHL; EBSCOHost; JSTOR; ProQuest; PubMed; SAGE Reference Online; ScienceDirect; SpringerLink; Taylor & Francis; and Wiley Online.” These databases would have likely added to the overall literature search in their academic rigor, aim, and biomedical scope. While OVID-MedLine, Scopus, and PsycINFO are excellent databases, the use of more databases would have added to the study sample size. Three databases just doesn’t seem to be enough for a more exhaustive comprehensive search.

Author Response

Nutrients 1573383

Due date for revisions: 20220131

We wish to thank the reviewers and editors for the careful review of our recently submitted manuscript, titled Food allergy education and management in schools: A scoping review on current practices & gaps. These comments have contributed to an improved manuscript, and to which we provide point-by-point responses below. The line numbers in the below correspond to those in the clean manuscript. We remain optimistic that these improvements are satisfactory to the reviewers and that the manuscript will be accepted for publication.

REVIEWER TWO

Comment 1

This was an interesting review that explores in-school management of food allergy, and perceived gaps or barriers in these management practices. The manuscript is certainly unique in its own right, however, there one major concern worth raising.

Response to Comment 1

Thank for your time and effort in reviewing our paper. We agree that this manuscript is unique as it is, as we believe, the first to report on an overview of food allergy management in schools.

Comment 2

My primary concern with this review involves the Methods section.

Firstly, the use of additional electronic academic databases would likely have added to the study in both complexity and sample size. If the scope of this study were expanded to use additional databases, more sources might have been identified and explored. For example, the following electronic databases could have been used for a more thorough and inclusive search;

“ArticleFirst; Biomed Central; BioOne; BIOSIS; CINAHL; EBSCOHost; JSTOR; ProQuest; PubMed; SAGE Reference Online; ScienceDirect; SpringerLink; Taylor & Francis; and Wiley Online.” These databases would have likely added to the overall literature search in their academic rigor, aim, and biomedical scope. While OVID-MedLine, Scopus, and PsycINFO are excellent databases, the use of more databases would have added to the study sample size. Three databases just doesn’t seem to be enough for a more exhaustive comprehensive search.

Response to Comment 2

Thank you for this suggestion. We agree having additional databases would have broadened the scope of the study. However, we wish to highlight the decision-making process behind the methods.

First, the use of three databases for searching was decided based on standard practice and recommendations guided by an expert research librarian and is in line with current literature (Arksey & O’Malley, 2005); PRISMA 2020 Statement (Page et al, 2020); The Joanna Briggs Institute Reviewers’ Manual, 2015). A minimum of three databases were also used in previously published studies from our group (Hildebrandt et al, 2021; Merrill et al, 2021) and in reviews published in Nutrients.

In addition, we believe that the three databases were sufficient in providing an inclusive search on our area of study. As noted in Figure 1 (line 368), the updated PRISMA flow sheet depicts the breakdown of types of articles screened for full-text which reflects the small percentage of articles that were pertinent for this specific schools focus. The study’s small sample size also further exhibits the gap in knowledge in this area, and the need for future work and research.

Lastly, we acknowledge that the method put forth in this paper may have introduced some reporting bias and may have reduced eligible studies. Thus, additions to the limitations section to reflect the same have been updated in the manuscript, as evidenced in lines 452-453.

References

Arksey H. & O’Malley L. Scoping studies: towards a methodological framework. International Journal of Social Research Methodology. 2005 Feb;8(1):19–32.

Hildebrand, H. V., Arias, A., Simons, E., Gerdts, J., Povolo, B., Rothney, J., & Protudjer, J. (2021). Adult and Pediatric Food Allergy to Chickpea, Pea, Lentil, and Lupine: A Scoping Review. The journal of allergy and clinical immunology. In practice, 9(1), 290–301.e2. https://doi.org/10.1016/j.jaip.2020.10.046

​Merrill KA, William TNN, Joyce KM, Roos LE, Protudjer JLP. Potential psychosocial impact of COVID-19 on children: a scoping review of pandemics & epidemics. J Global Health Rep.

Page M.J., McKenzie J.E., Bossuyt P.M., Boutron I., Hoffmann T.C., Mulrow C.D., et al. The PRISMA 2020 statement: an updated guideline for reporting systematic reviews. BMJ. 2021 Mar 29;n71.

Peters, M.D., Godfrey, C.M., McInerney, P., Soares, C.B., Khalil, H., & Parker, D. (2015). The Joanna Briggs Institute reviewers' manual 2015: methodology for JBI scoping reviews.

Round 2

Reviewer 2 Report

thank you for addressing these comments!